

# Cyclonic intensity study using sea level pressure estimations from Ocensat-II scatterometer winds over Bay of Bengal during 2013

Purna Chand C[1], Venkateswara Rao M[1], Prasad K V S R[2], and Rao K H[1]

[1]National Remote Sensing Centre, Hyderabad. India

[2]Dept. of Meteorology and Oceanography, Andhra University, Visakhapatnam, India

*Correspondence to*: Purna Chand C (purnachandch@gmail.com)

**Abstract**. Sea level pressure (SLP) fields prevailing over Bay of Bengal (BoB) during Phailin, Lehar and Madi cyclones (2013) were estimated using University of Washington Planetary Boundary Layer (UWPBL) model with Oceansat-II

Scatterometer (OSCAT) sea surface winds as an input parameter. Model performance in estimating SLP variations during cyclonic periods was investigated by comparison against cyclone reports of Indian Meteorological Department (IMD) and in-situ measurements. Pressure drop as per IMD reports were observed to be higher than model estimates, primarily due to limitations in scatterometer wind data. However, model retrieved pressure fields compare well against buoy measurements, with a bias of 0.0042, -0.1279 and -0.4406 observed for Phailin, Lehar and Madi cyclones respectively. The contrast in pressure drops between

model estimates and IMD reports was investigated by comparing scatterometer winds against IMD reported maximum sustainable winds.

## 1 Introduction

Tropical cyclones develop over the North Indian Ocean (NIO) mostly during the pre and post monsoon seasons. Studies on cyclones occurring in the NIO region for the period spanning 1877-1998 have revealed a slightly decreasing, but insignificant, trend of -0.8/100 years in tropical cyclone frequency (Singh et al., 2000, 2001 and Rao et al., 2001). The Bay of Bengal region usually records the highest frequency of cyclones in the NIO. However, it is not unusual for a storm to move across southern India and re-intensify over the Arabian Sea, especially in October month during which the cyclone occurrence frequency is

typically at its highest. Cyclones inflict devastation in the form of torrential rainfall, flooding, storm surges and gales causing death and destruction to property. Storm surges, which are an abnormal rise in water level beyond the predicted astronomical tides, typically vary between 0.3 – 8m along the Indian coastline (Shaji et al., 2013), with a maximum of 12.5m recorded at Backerganj, Bangladesh in 1876 (Flemming et al., 2006). It is estimated that cyclones have claimed over 9,00,000 victims over the past 30 years in Bangladesh alone.


Cyclones are characterized by closed iso-bars and high-velocity winds. The rise in sea levels along the coast due to cyclonic influence, storm tides, depends on a combination of meteorological forcing's and non-meteorological factors such as astronomical tide, depth of the bay and local topography. Remote sensing satellites are currently incapable of directly capturing storm surge and pressure fields over the sea surface. Sea surface winds retrieved from scatterometer data offer an opportunity to

obtain sea level pressure fields as demonstrated by several researchers (Brown and Levy 1986; Brown and Zeng 1994; Harlan and O'Brien 1986; Hsu and Liu 1996; Hsu et al., 1997; Zierden et al., 2000). Patoux et al., (2003) applied inverse model to estimate sea surface pressure gradients and fields in the tropics, using surface wind fields with a PBL model (Stevens et al.,



2002). The UWPBL model (Patoux 2004) was developed by integrating the PBL model with a mid-latitude model to provide continuous global pressure fields. Patoux et al., (2008) have established the root mean square difference between QuikSCAT based surface pressure fields with European Centre for Medium-Range Weather Forecasts (ECMWF) surface pressure fields is on the order of 1 hPa in tropics and 2 hPa in the mid-latitudes, this is consistent with the results reported by Chelton and

Freiliche (2005).

A wavelet based method is described to blend scatterometer derived surface pressure swaths into ECMWF analysis by Patoux et al., (2009), in his results modified pressure fields are used to identify and track mid-latitude cyclones over the southern ocean, as well as to estimate the depth and size of cyclones at each stage of their period during July 1999 to June 2006. Identified more

low pressure centers with the modified pressure fields, in particular incipient lows captured earlier than ECMWF and more mesoscale cyclones (less than 4 dyas life span). Lei- Ming et al., (2010) had introduced a new scheme, named Vortex Initialization with the Assimilation of Retrieved Variables, is demonitrated to improve the initialization of regional numerical model for predication of Tropical Cyclone. Maximum wind speed and low sea level pressure fields retrieved from the QuikSCAT data obtained using the modified UWPBL model  given good agreement with the observations relative to derived

parameters from the analysis of  National Center for Environmental Prediction global model (NCEP). Purna Chand et al., (2014) demonstrated the estimation of OSCAT-derived pressure fields of the cyclone Nilam well, both in track and intensity, compared to the ECMWF re-analysis fields.

In this study, we have employed the UWPBL model to study the intensity of three severe cyclonic storms, Phailin, Lehar and

Madi. The year 2013 in which these storms occurred, recorded the highest cyclonic activity over the Bay of Bengal observed in recent years, with the occurrence of 10 depressions, 3 of which evolved into severe cyclonic storms. The damage due to Phailin cyclone alone is nearly 50% of the total damage caused by all other cyclonic events. We have made an attempt to study the accuracy and intensity of sea level pressure fields from UWPBL model during cyclonic periods.

**2 Data and Methodology**

Sea surface winds data have been obtained from OSCAT scatterometer on-board Oceansat-II platform. In-situ measurements of sea surface winds from moored buoys have been obtained from Indian National Centre for Ocean Information Services (INCOIS). Scatterometer instrument is basically a radar operating in the Ku-band (13.515 GHz) with two pencil beams and

measures backscatter coefficient ($\sigma_o$) in four azimuth angles, enabling the estimation of wind vectors in 50 x 50 km cells. Chakraborty et al. (2013) reported that OSCAT wind speeds compare favorably against winds from ECMWF (European Centre for Medium-Range Weather Forecasts) and NCEP (National Centers for Environmental Prediction). Pressure fields derived from OSCAT winds have been utilized to study Nilam cyclone (PurnaChand et al., 2014).

In the present study, we have estimated sea level pressure fields during Phailin (08 - 14 October 2013), Lehar (23 – 28 November

2013) and Madi (06 – 13 December 2013) cyclonic storms (cyclone tracks illustrated in Fig. 1) using UWPBL model (Patoux et al., 2004) with OSCAT L2B winds. Surface pressure gradients generated by the model, using 2 layer similarity and mixed layer models for the mid-latitudes and tropical regions respectively, are blended for overlapping latitudes using least square optimization method. Absolute pressure fields are generated from the pressure gradient by supplying initial or first guess values. Boundary layer conditions are computed by the model from 925hPa background winds, substituted from NCEP/NCAR

climatological re-analysis (Kalnay et al., 1996) for the present study. Retrieval of sea level pressure fields from UWPBL model





has been discussed in further detail in PurnaChand et al., 2014. Model estimated pressure fields and scatterometer winds have been compared against IMD cyclone centre pressures and maximum sustainable wind speeds respectively.

## 3 Results and Discussions

Sea level pressure fields estimated by UWPBL model using OSCAT winds during Phailin, Lehar and Madi cyclones are presented in Fig.2. Phailin cyclone originated from a depression which developed on 8[th] October 2013 and intensified into a severe cyclonic storm by 10[th] October 2013, as depicted by the pressure fields, with cyclone eye pressure less than 1000hPa. It further intensified with an estimated central pressure of 994hPa and crossed Orissa coast near Gopalpur on 12[th] October 2013. Lehar cyclone developed from a depression near Andaman Islands on 23[rd] November 2013 and intensified into a severe cyclonic storm by evening of 25[th] November 2013. It later crossed the coast near Machilipatnam, Andhra Pradesh state. Similarly, pressure fields reveal formation of a depression on 6[th] December, 2013 which intensified to a very severe cyclonic storm by 8[th] midnight/9[th] early morning, followed by decrease in intensity from the next day and landfall at Vedaranyam, Tamil Nadu state on afternoon of 12[th] December, 2013.

Cyclone eye pressure values estimated by the UWPBL model along the track of all three cyclones have been compared against IMD reported pressures (Fig. 3a-c). We observe that Phailin cyclone is the most intense as indicated by pressure drop by both IMD reports and model estimates. Model estimates cyclone eye pressure at peak intensity to be 994 hPa while IMD reports indicate it is much lower at 940 hPa. However, buoy measurements compare favorably with model estimates.

A closer examination of Fig.3 reveals relatively good agreement between IMD and model estimated pressure fields during formation phase of cyclone, followed by an increasing overestimation of cyclone eye pressure by model as the cyclone intensifies. This may possibly be due to saturation of OSCAT winds beyond 24m/s. However, wind speeds greater than 24m/s are frequently observed during the peak phase of severe cyclonic storm. This is amply demonstrated in the difference between model estimates and IMD reported pressures, which is highest for Phailin, the most severe cyclone considered in the present study.

The accuracy of scatterometer winds was investigated by comparing OSCAT winds against IMD estimates of maximum sustainable winds, revealing large differences. Deviations between scatterometer and IMD winds (Fig.4) also exhibit similar pattern to pressure deviations as illustrated in Fig.3.

Comparison of model derived pressure fields with buoy measurements is illustrated in Fig.5 and statistical details are listed in Table 1. Buoy observed pressure values were obtained from 9 buoy locations (Fig. 1) for all three cyclones, with the lowest bias between buoy and model pressures observed for Phailin while the highest correlation was observed for Lehar. Statistical parameters computed include Root Mean Square Deviation (RMSD) to estimate percentage of expected error for the parameter and Standard Deviation Ratio (SDR), the ratio between standard deviation of error and standard deviation of in-situ observations, to analyze whether data is close to observed or in-situ observations.

## 4 Conclusions

Sea level pressure fields were estimated using UWPBL model during Phailin, Lehar and Madi, three very severe cyclonic storms which occurred in 2013. The pressure drop between formation phase and peak phase of cyclone is the highest for Phailin, indicating it is more severe than the other two cyclones.



Pressure fields derived from UWPBL model are observed to be close to both IMD reported pressure fields and buoy measurements when OSCAT scatterometer winds, given as input to model, are within the range of sensor limitations (4-24 m/s). Increasing divergence between model estimates and IMD pressures is noticed where IMD maximum sustainable winds exceed 24m/s, suggesting that the deviation between model estimated and IMD reported pressures may be due to limitations associated

with scatterometer based sea surface wind measurements. Model estimates are observed to agree well with buoy measurements for all three cyclones. We conclude that UWPBL model is suitable to study sea surface pressure variations, however, the limitations associated with utilizing scatterometer winds as model input should be considered before using the model to study pressure variations during cyclonic periods.

## 5 Acknowledgements

This work was carried out at NRSC, as a part of Oceansat-II utilisation and National Information System for Climate and Environmental Studies (NICES) program. The OSCAT wind data obtained from NRSC/ISRO. The in-situ pressure data collected from INCOIS and cyclone detailed data collected from IMD. The authors thankful to J. Patoux, for his support rendered during the implementation of UWPBL model at NRSC.

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


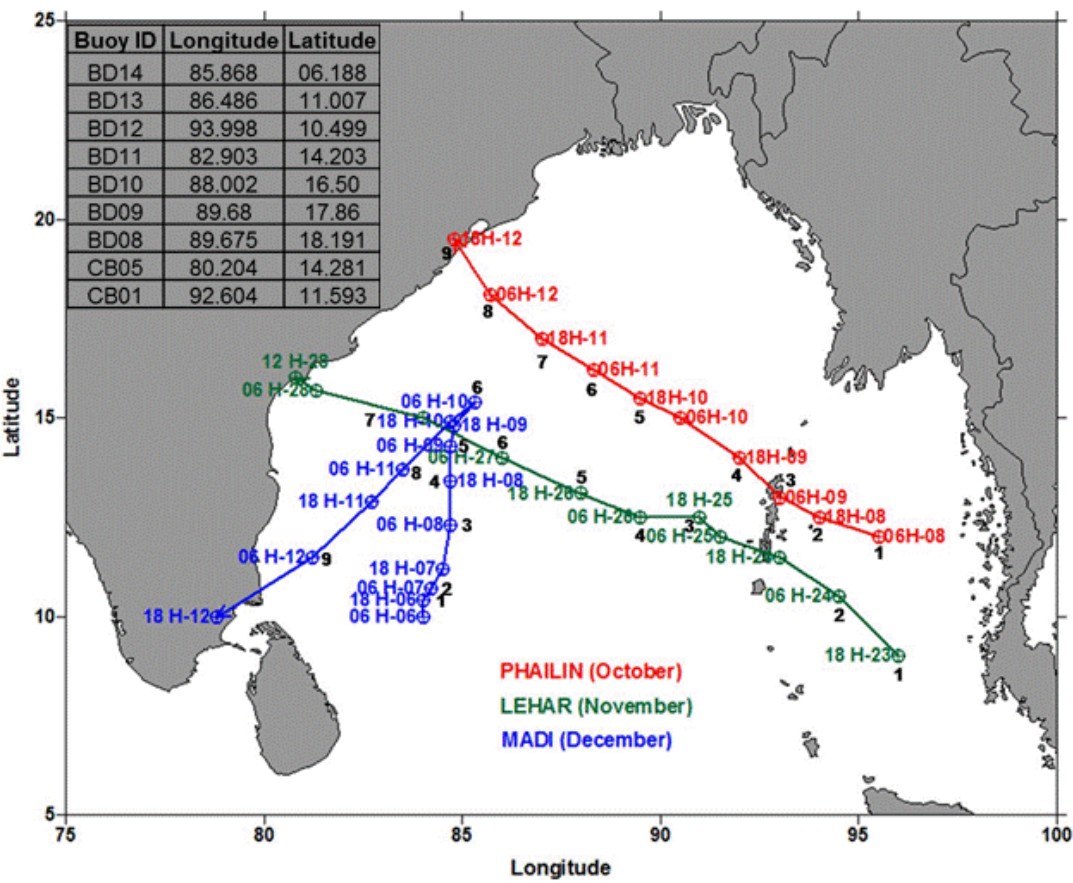

**Figure 1**. Very Severe Cyclonic Storm tracks during 2013 along with location numbers and Buoy locations used for validation.





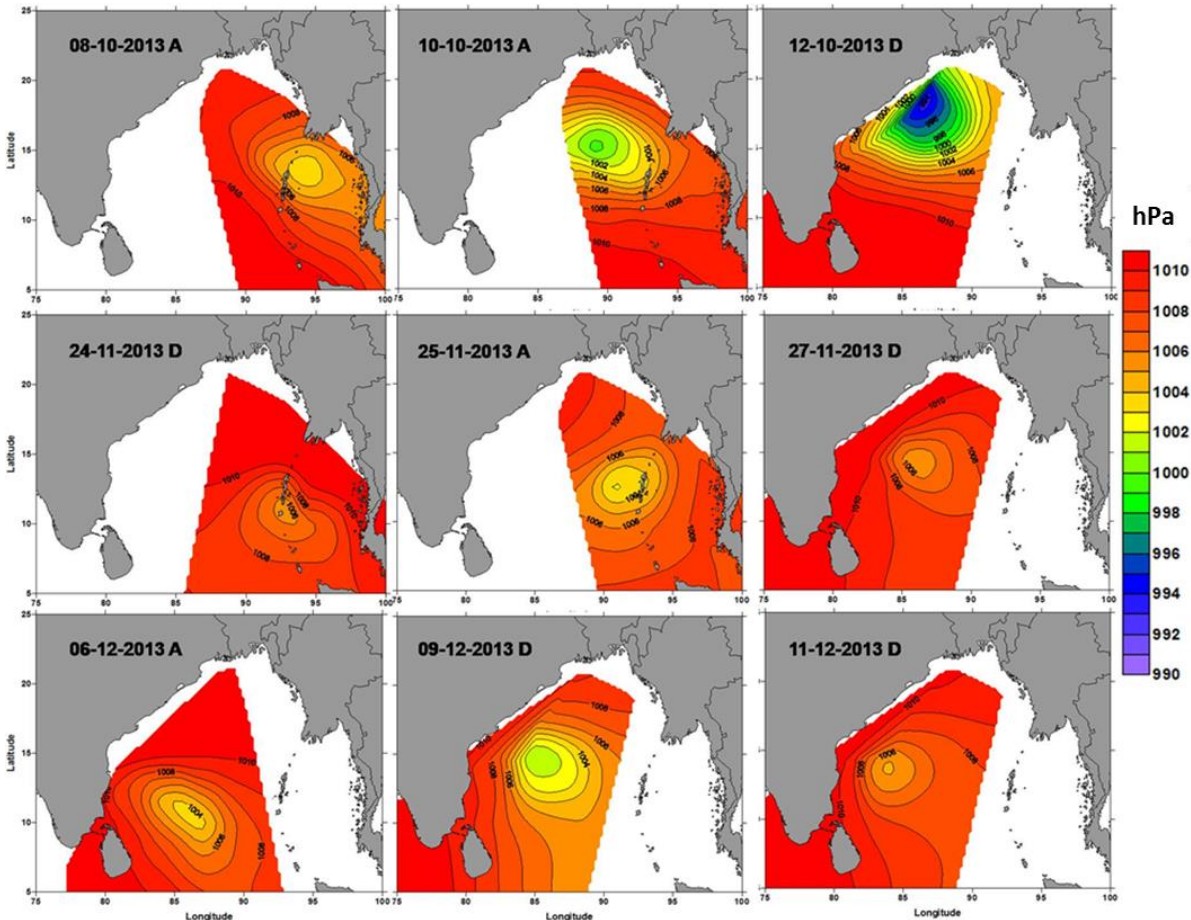

**Figure 2**. Pressure fields retrieved from OSCAT winds using UWPBL model during Phailin, Lehar and Madi cyclones (after the date, A- Ascending Pass, D – Descending Pass).




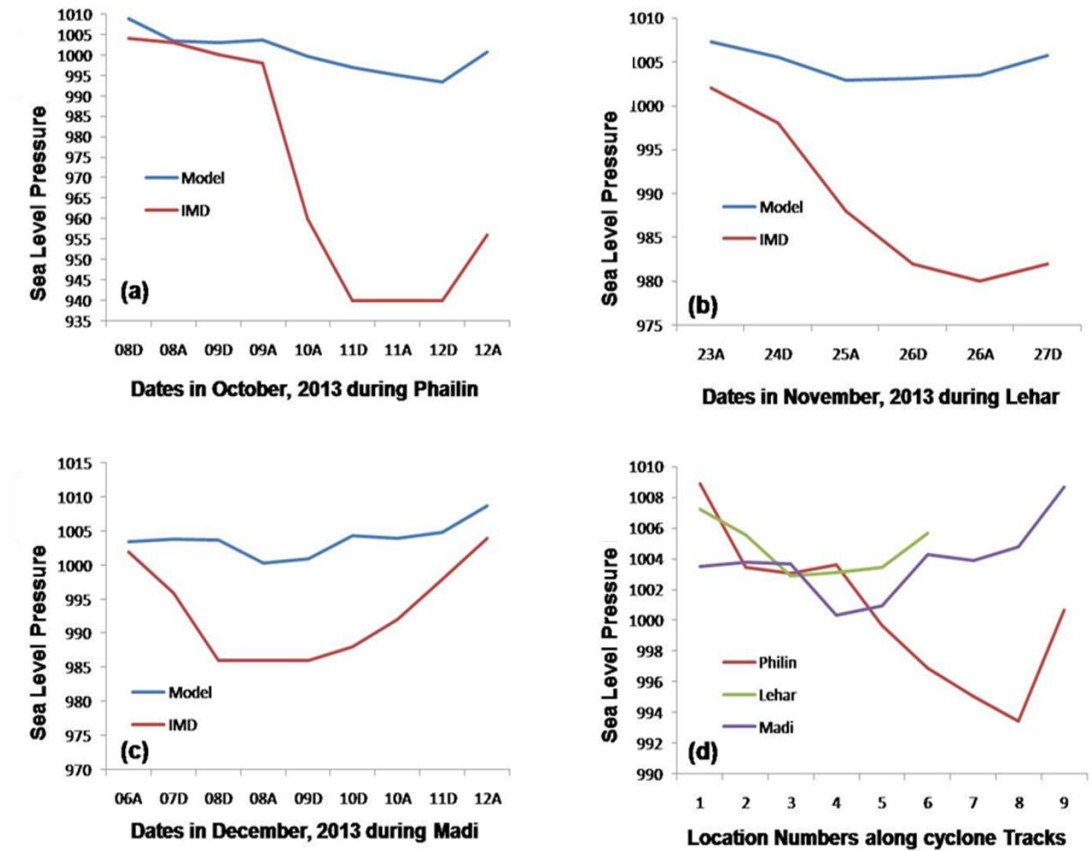

**Figure 3**. Pressure (hPa) variations along the cyclone track (a) Phailin, (b) Lehar, (c) Madi and (d) for all the three according to the location numbers mentioned in figure 1.



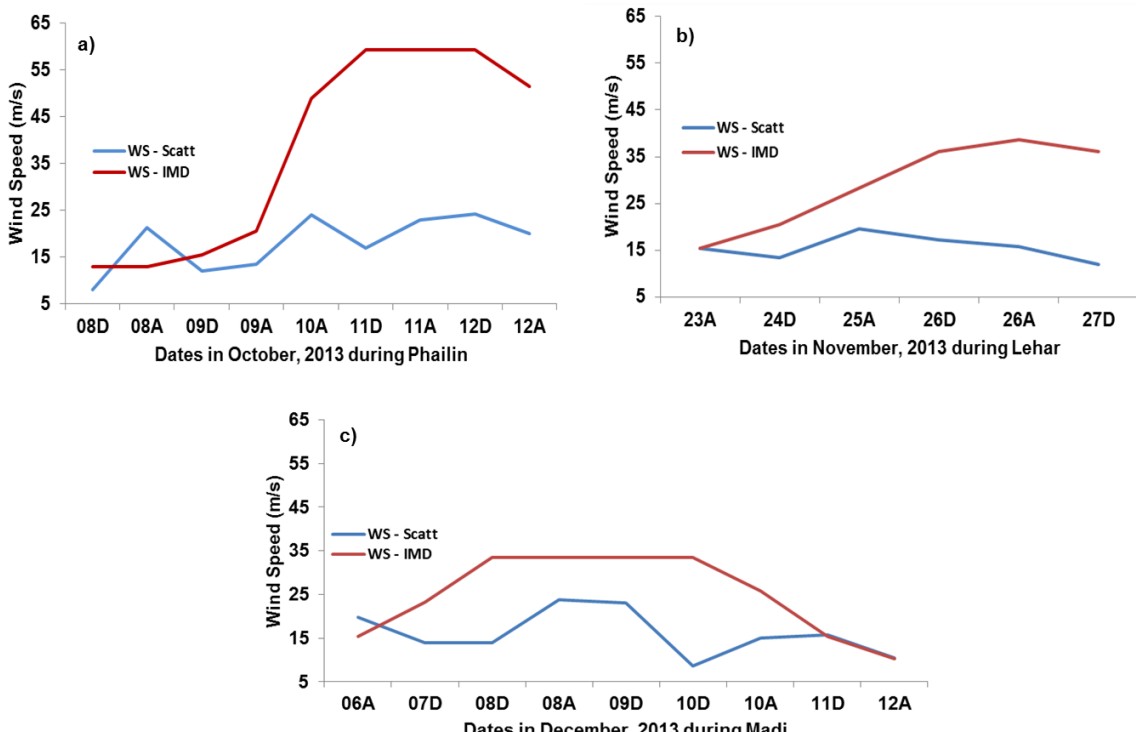

**Figure 4**. Comparison of OSCAT winds and maximum sustainable winds from IMD along the track (a) Phailin, (b) Lehar and (c) Madi.





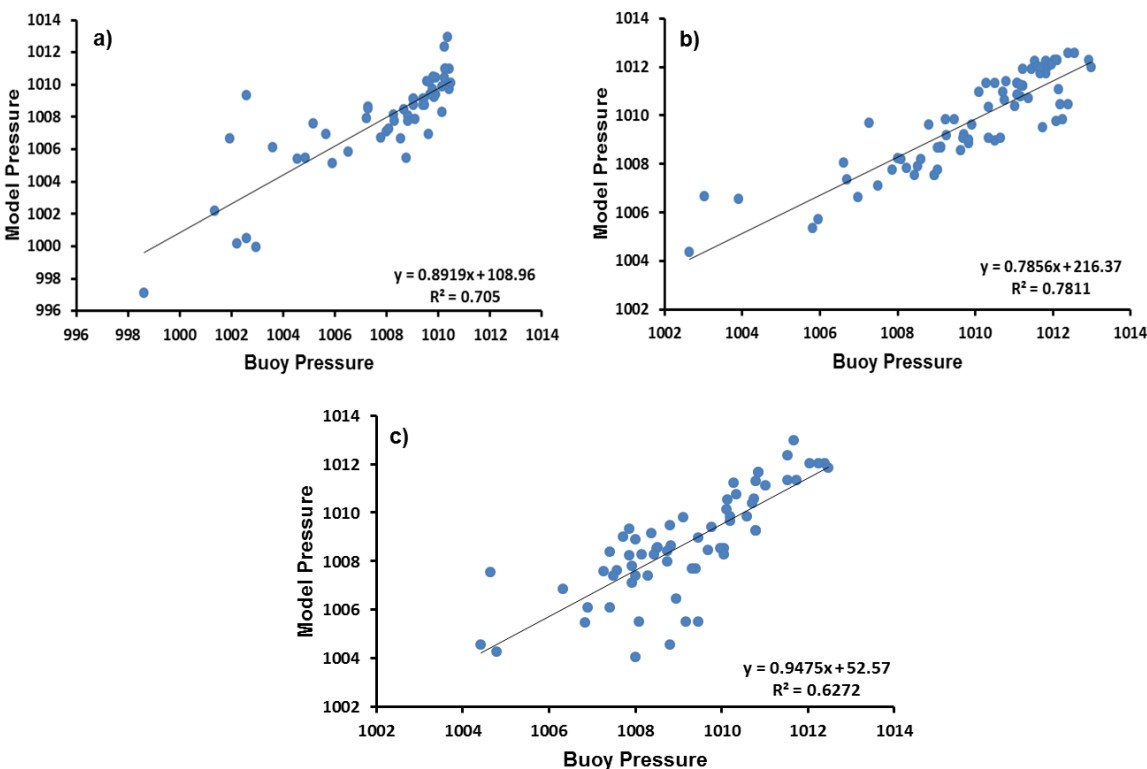

**Figure 5**. The Scatter between the in-situ and model derived pressure (hPa) during the (a) Phailin, (b) Lehar and (c) Madi cyclonic periods.

**Table 1**. Statistical analysis of the comparison between model derived and in-situ observation during the three cyclonic periods in 2013.

| Cyclone | No. of points collocated | Bias | R C | C C | RMSD | S D R | S I |
|---|---|---|---|---|---|---|---|
| **Phailin** | 56 | 0.0042 | 0.705 | 0.839 | 1.668 | 0.587 | 0.00165 |
| **Lehar** | 77 | -0.1279 | 0.781 | 0.883 | 1.034 | 0.468 | 0.00102 |
| **Madi** | 66 | -0.4406 | 0.627 | 0.791 | 1.381 | 0.732 | 0.00136 |
| R C – Regression Coefficient, C C – Correlation Coefficient, RMSD – Root Mean Square Deviation, S D R – Standard Deviation Ratio, S I – Scatter Index. | | | | | | | |