# Peer review of "Cyclonic intensity study using sea level pressure estimations from Ocensat-II scatterometer winds over Bay of Bengal during 2013"

_Natural Hazards and Earth System Sciences, 2017_

## Referee Comment (RC1) · R. Foster (Referee) · 26 Jul 2017

Title: Cyclonic intensity study using sea level pressure estimations form Oceansat-II scatterometer winds over Bay of Bengal during 2013

Authors: C Purna Chand, M.V. Rao, K.V.S.R. Prasad, and K.H.Rao

Recommend: Reject in its present form. I am willing to review a re-submitted manuscript.

The paper is premature and will leave a false impression in the literature that OceanSAT is incapable of retrieving sea-level pressure in tropical cyclones. This reviewer has

retrieved good quality SLP using OceanSAT scatterometer wind data and predicts that a successful system could be developed, which has the potential to benefit to the Indian meteorological community.

Synopsis: This paper describes the application of the generic "UW" sea-level pressure retrieval model to OceanSAT-II observations of three tropical cyclones (TCs). Very poor results are obtained; in one case the MSLP derived from the scatterometer is off by $\sim$54 mb when compared to the best track estimate.

The reasons for this discrepancy are obvious and have been discussed at various meetings. In summary: (1) the standard wind speed processing of the Ku-band scatterometers such as QuikSCAT and OceanSAT severely under-estimate the wind speeds beyond $\sim$25 m/s; (2) the wind direction retrievals in TC cores are biased, which can be easily seen in the derived TC shapes of Fig. 2; (3) the two-layer PBL model employed in the generic UW SLP code is a poor representation of the nonlinear mean flow dynamics of the TC boundary layer.

With reference to (1), if the scatterometer wind speeds are biased low (very low in the TC case), even a perfect PBL model will consequently produce low (or very low) estimates of the corresponding surface pressure gradient magnitudes.

With reference to (2), an important way to think about the directional errors is to consider how a mis-aligned wind vector will alias the azimuthal flow (primary circulation) into the radial flow (secondary circulation). Radial pressure gradients are much stronger than azimuthal pressure gradients. The strong radial pressure gradients are largely balanced by the nonlinear mean flow dynamics that are first-order in the TC PBL, but small in "normal" PBLs – see below.

Even if the pressure gradient magnitudes were correctly estimated by the surface winds and PBL model, systematic errors in the surface wind direction will pass through to errors in the corresponding pressure gradient orientations. Since the SLP code effectively integrates through the pressure gradients, the resulting field will have the wrong

shape. (Random direction errors, if small enough, can be tolerated.)

With respect to (3), the TC PBL dynamics are quite different from the the basic PBL dynamics described in the Stevens et al. (2002) model. The important feature of the Stevens et al. PBL model is that it maintains a three-way force balance (ala mid-latitude Ekman layers) by substituting PBL-top momentum entrainment for the Coriolis force, which approaches zero at low latitudes. However, its scaling is inappropriate for the intense, swirling TC PBL mean flow.

Of key importance is that standard PBL scaling shows that only the nonlinear terms involving turbulent perturbations (Reynolds stresses) are of leading order in the mean flow equations. With appropriate substitutions and sign changes, the zonal and meridional PBL equations are the same.

In contrast, the TC PBL scaling shows that while the Reynolds stresses remain important, the nonlinear terms involving mean flow velocities are also of leading order. Inherently in the TC PBL, the equation describing the radial momentum balance is different from that describing the azimuthal momentum balance. Without the nonlinear mean flow dynamics, the strong radial pressure gradients would be be improperly balanced by other terms in the PBL model. This leads to errors in the estimated pressure gradients and resulting SLP fields.

It is possible to recover accurate sea-level pressure fields from remotely sensed surface wind vectors in TC conditions once these three issues are addressed. In the case of scatterometer winds, the standard model function-based wind retrievals should be replaced with TC-specific wind retrievals (e.g. Stiles et al. 2013). Secondly, the wind direction retrievals require specific filtering to improve their accuracy. Thirdly, the PBL model incorporated in the SLP retrieval code must be replaced by one that makes a better approximation to the TC PBL dynamics.

Using high-quality SAR surface winds, RMS errors in the SLP are less than 4 mb when compared to aircraft drop sonde observations. Such processing has also been

Interactive
comment
performed using QuikSCAT surface wind fields from the Stiles et al. processing. Compared to aircraft drop sondes, RMS errors of 5 to 6 mb are found. OceanSAT is fairly similar to QuikSCAT. So, I would expect that comparable SLP processing is possible given sufficient care. Preliminary results with OceanSAT winds have been promising.

I have attached a few plots showing an SLP retrieval of Hurricane Earl observed by OceanSAT on September 1, 2010. A TC-specific wind speed retrieval method was used. The PBL model in the UW SLP code was replaced by one that makes a better approximation to TC scaling. This example shows the residual importance of filtering out the error in the OceanSAT wind directions on the retrieved SLP fields leaving the wind speeds unchanged.

――――――――――――――――――――

[Figure]

**Fig. 1.** OceanSAT wind directions

[Figure]

**Fig. 2.** Improved wind directions

**Fig. 3.** Drop sonde pressure differences using OceanSAT wind directions

Fig. 4. Drop sonde pressure differences using improved wind directions

---

## Referee Comment (RC2) · Anonymous Referee #2 · 26 Jul 2017

Paper: Cyclonic intensity study using sea level pressure estimations from Ocensat-II scatterometer winds over Bay of Bengal during 2013 Authoer: C. Purna Chand, M. Venkateswara Rao, K. V. S. R . Prasad, and K. H. Rao

Summary: This paper uses a PBL model from the University of Washington with Oceansat-II scatterometer winds as an input to derive a surface pressure field for three tropical cyclones in the Bay of Bengal during 2013.

General comments: The approach here is flawed in that the sensor being used (Oceansat-II) is incapable of resolving the winds in the core of most tropical cyclones and thus any attempt at retrieving the pressure field will result in severe under-

estimation of both the MSLP and the tight pressure gradients without substantial pre-processing of the data. The authors note these deficiencies but I believe that additional work needs to be done to address the shortcomings of the approach. There a number of grammatical errors that I noted that should be corrected if the paper is re-submitted. Is it Ocensat or OceanSat?

Abstract: "Pressure drop as per IMD reports were observed to be higher than model..." Much higher in the cases of both Phailin (> 50 hPa!) and Lehar (> 20 hPa).

"However, the model retrieved pressure fields compare well against buoy measurements" This implies that buoy observations are acting as ground truth including the TC inner core region and thus the model does well. This is very misleading. The authors do not note where the buoys are located relative to the TC center. Looking over figure 5 it looks like none of these observations are near the TC inner core?

Paper Body: Pages 2 section 5 - "Identified more low pressure ..." (grammar)

Page 2 section 10 - "4 dyas life span" - (spelling)

Page 2 section 30 - "OSCAT winds compare favorably against ECMWF and NCEP". Comparing "observations" to model fields notwithstanding there is plenty of literature that looks at the particular challenges of using scatterometer winds in tropical cyclones. The authors may be able to use the approach of Stiles et al (2014) to improve the quality of the OSCAT data in order to retrieve winds > 25 m/s. See:

Stiles, B. W., R. E. Danielson, W. L. Poulsen, M. J. Brennan, S. Hristova-Veleva, T-P. Shen, and A. G. Fore, 2014: Optimized tropical cyclone winds from QuikSCAT: A neural network approach. IEEE Trans. Geosci. Remote Sens., 52, 7418–7434.

Page 3 section 5: "Phailin cyclone ... eye pressure less than 1000 hPa". It is confusing in this discussion of the intensity whether the authors are referring to IMD pressures or the model pressures?

Page 3 section 5: No discussion here regarding the asymmetries in the pressure fields

in Figure 2? Is this an issue with processing of the ambiguities?

Page 3 section 15: "Buoy measurements compare favorably to model estimates". This is misleading as it suggests that while IMD estimates are a poor match to the model in-situ buoy observations verify the model is skillful. There is no discussion here to note that the buoys are not located near the TC center. Plots showing the buoy locations relative to the TC at the time of the OSCAT pass would clarify this.

The plots in figure 3 clearly show the under-estimation of pressure of the model compared to the IMD estimates. So it is not clear to me how this model would be applied? Certainly it would not be used to estimate intensity?

Figures 5 and table 1: Again it needs to be made clear what exactly is being compared here lest the readers come away with conclusion that the model can accurately estimate the MSLP of even the most intense TCs.

Summary: I appreciate what the authors are attempting to do here. Estimating TC intensity in the absence of observational data such as aircraft reconnaissance is a challenge. However OSCAT (and other scatterometers) can at best only be used to estimate the intensity in weaker storms with an inner core that has sufficient diameter to be resolved by the instrument or wind structure information such as the radius of gales. Additional skill may be obtained using the neural network approach of Stiles but if must be developed specifically for OSCAT (though QSCAT is very similar so it should work).

---

## Author Comment (AC1) · 9 Oct 2017

Dear Dr. Ralph Foster (Referee),

Thank you very much for your valuable suggestions and comments. We will incorporate the resolutions of comments each point by point. In this present study we are estimated the pressure fields from OSCAT winds using UWPBL model and compared with the in-situ observation these results are well compared. But the cyclone center values are having lag when compared with the IMD analysis. As well as in revised manuscript, we compared ECMWF re-analysis mean sea level pressure with UWPBL model estimations. In this study the pressure values are well matching with the UWPBL model estimated pressure fields in the center of the cyclone corresponding to the best track data.

With reference to (1), if the scatterometer wind speeds are biased low (very low in the TC case), even a perfect PBL model will consequently produce low (or very low) estimates of the corresponding surface pressure gradient magnitudes.

Authors Response: Yes, we too agree with your observation and experience.

With reference to (2), an important way to think about the directional errors is to consider how a misaligned wind vector will alias the azimuthal flow (primary circulation) into the radial flow (secondary circulation). Radial pressure gradients are much stronger than azimuthal pressure gradients. The strong radial pressure gradients are largely balanced by the nonlinear mean flow dynamics that are first-order in the TC PBL, but small in "normal" PBLs – see below.

Even if the pressure gradient magnitudes were correctly estimated by the surface winds and PBL model, systematic errors in the surface wind direction will pass through to errors in the corresponding pressure gradient orientations. Since the SLP code effectively integrates through the pressure gradients, the resulting field will have the wrong shape. (Random direction errors, if small enough, can be tolerated.)

Authors Response: What you said is correct in the consideration of directional errors. As per the OSCAT mission goal of 2 $ms^{-1}$ wind speed accuracy and $20°$ in direction, in the range of $4 - 24$ $ms^{-1}$ wind speed is maintained from the OSCAT these wind speed and directional accuracies are well compared with ECMWF and NCEP analysis products by Chakraborty *et al.*, 2013.

With respective to your 3rd point, we added filters to remove the directional errors to address without changing the wind speed. As you mentioned in the last point "Using high-quality SAR surface winds, RMS errors in the SLP are less than 4 mb when compared to aircraft drop sonde observations", in this study we observed the RMS errors are less than 1.7 mb with the in-situ observation. We will investigate why lag is high between IMD analysis and UWPBL estimations in further studies.

---

## Author Comment (AC2) · 9 Oct 2017

Dear Referee,

Thank you very much for your valuable suggestions and comments. We incorporated the resolutions of comments each point by point.

General comments: The approach here is flawed in that the sensor being used (Oceansat-II) is incapable of resolving the winds in the core of most tropical cyclones and thus any attempt at retrieving the pressure field will result in severe under-estimation of both the MSLP and the tight pressure gradients without substantial preprocessing of the data. The authors note these deficiencies but I believe that additional work needs to be done to address the shortcomings of the approach. There a number of grammatical errors that I noted that should be corrected if the paper is re-submitted. Is it Ocensat or OceanSat?

Authors Response: Thank you very much for your observations in the manuscript, some additional work has been carried out do address the pressure difference in center of the cyclone, which work was presented in the revised manuscript. Grammatical errors will be corrected as you mentioned in the revised manuscript. Whatever we used that sensor is OCEANSAT, we corrected this one also in respective place.

Abstract: "Pressure drop as per IMD reports were observed to be higher than model..." Much higher in the cases of both Phailin (> 50 hPa!) and Lehar (> 20 hPa).

"However, the model retrieved pressure fields compare well against buoy measurements" This implies that buoy observations are acting as ground truth including the TC inner core region and thus the model does well. This is very misleading. The authors do not note where the buoys are located relative to the TC center. Looking over figure 5 it looks like none of these observations are near the TC inner core?

Authors Response: We mentioned the buoy locations in the figure 1 in tabular form. As you mentioned, there is no moored buoys are available near to the cyclone center during these three cyclone event times. To address this, we compared those locations with the WCMWF re-analysis pressure values. These values are well matching with the estimated values.

Paper Body: Pages 2 section 5 - "Identified more low pressure ..." (grammar)

Authors Response: Modified accordingly

Page 2 section 10 - "4 dyas life span" - (spelling)

Authors Response: Corrected

Page 2 section 30 - "OSCAT winds compare favorably against ECMWF and NCEP". Comparing "observations" to model fields notwithstanding there is plenty of literature that looks at the particular challenges of using scatterometer winds in tropical cyclones. The authors may be able

to use the approach of Stiles et al (2014) to improve the quality of the OSCAT data in order to retrieve winds > 25 m/s.

Authors Response: What you said is correct, but in this study we are not retrieving the winds, we focused on estimations of pressure fields from the available scatterometer wind data. This is very good point for further studies to improve the quality of the estimations of pressure as well as the wind fields.

Page 3 section 5: "Phailin cyclone ... eye pressure less than 1000 hPa". It is confusing in this discussion of the intensity whether the authors are referring to IMD pressures or the model pressures?

Authors Response: Here we are referring the model estimated pressure not IMD

Page 3 section 5: No discussion here regarding the asymmetries in the pressure fields in Figure 2? Is this an issue with processing of the ambiguities?

Authors Response: Thank you very much for the observation, there is no processing ambiguities. We are not mentioned any asymmetries due to those are different cyclones.

Page 3 section 15: "Buoy measurements compare favorably to model estimates". This is misleading as it suggests that while IMD estimates are a poor match to the model in-situ buoy observations verify the model is skillful. There is no discussion here to note that the buoys are not located near the TC center. Plots showing the buoy locations relative to the TC at the time of the OSCAT pass would clarify this.

Authors Response: Modified accordingly and clarified in revised manuscript.

The plots in figure 3 clearly show the under-estimation of pressure of the model compared to the IMD estimates. So it is not clear to me how this model would be applied? Certainly it would not be used to estimate intensity?

Authors Response: Here we compared the Sea Level Pressure values with the IMD pressure values, in revised manuscript we added ECMWF pressure values also.

Figures 5 and table 1: Again it needs to be made clear what exactly is being compared here lest the readers come away with conclusion that the model can accurately estimate the MSLP of even the most intense TCs.

Authors Response: Modified accordingly in revised manuscript.